# Direct catalytic hydrogenation of $CO_2$ to formate over a Schiff-base-mediated gold nanocatalyst

Qinggang Liu[1,2], Xiaofeng Yang[2], Lin Li[2], Shu Miao[2], Yong Li[3], Yanqin Li[1], Xinkui Wang[1], Yanqiang Huang[2] & Tao Zhang[2]

Catalytic transformation of $CO_2$ to formate is generally realized through bicarbonate hydrogenation in an alkaline environment, while it suffers from a thermodynamic sink due to the considerable thermodynamic stability of the bicarbonate intermediate. Here, we devise a route for the direct catalytic conversion of $CO_2$ over a Schiff-base-modified gold nanocatalyst that is comparable to the fastest known nanocatalysts, with a turnover number (TON) of up to 14,470 over 12 h at 90 °C. Theoretical calculations and spectral analysis results demonstrate that the activation of $CO_2$ can be achieved through a weakly bonded carbamate zwitterion intermediate derived from a simple Lewis base adduct of $CO_2$. However, this can only occur with a hydrogen lacking Lewis base center in a polar solvent. This finding offers a promising avenue for the direct activation of $CO_2$ and is likely to have considerable implications in the fields of $CO_2$ conversion and gold catalytic chemistry.

[1] State Key Laboratory of Fine Chemicals, School of Chemistry, Dalian University of Technology, Dalian 116024, China. [2] Dalian Institute of Chemical Physics, Collaborative Innovation Center of Chemistry for Energy Materials, Chinese Academy of Sciences, Dalian 116023, China. [3] Institute of Applied and Physical Chemistry and Center for Environmental Research and Sustainable Technology, University of Bremen, 28359 Bremen, Germany. Correspondence and requests for materials should be addressed to X.Y. (email: yangxf2003@dicp.ac.cn) or to X.W. (email: wangxinkui@dlut.edu.cn) or to Y.H. (email: yqhuang@dicp.ac.cn)

Catalytic hydrogenation of $CO_2$ has attracted much attention in recent years, because it not only serves to mitigate the problem of anthropogenic emissions of $CO_2$ but also provides a feasible avenue for carbon recycling and hydrogen energy conversion or storage[1-4]. Among the various hydrogenation products of $CO_2$, formic acid, which presents as formate in the practical production for the thermodynamic shift of equilibrium forward, is one of the most attractive, owing to its direct employment as a feedstock chemical or hydrogen source for fuel cells[5]. In addition, the conversion of $CO_2$ to formic acid is believed to be the first and indispensable step in the reduction of $CO_2$ to other chemicals or fuels[6], such as methanol, methane, or other hydrocarbons, and a fundamental understanding of this process is thus essential for C1 chemistry[7].

The activation of $CO_2$ is often challenging, owing to the high thermodynamic stability of this molecule[8,9]. Considering that $CO_2$ acts as an electrophile or Lewis acid[10], $CO_2$ is generally activated using an electron donor or base. For example, high $CO_2$ conversions have been achieved using homogeneous catalysts containing electron-donating ligands on the metal active sites[11,12]. However, the catalytic activity in homogeneous catalysis is very sensitive to the ligand used and the ligands are expensive and leachable, which limits their wide-spread application. Without ligand promotion in a heterogeneous catalytic system, $CO_2$ is always activated in a bent conformation by the interaction between the dissolved base with $CO_2$[13]. In such a process, $CO_2$ chemically reacts with an aqueous base, such as KOH or NaOH, to give bicarbonates ($HCO_3^-$), which serve as the real precursors for further hydrogenation to formate[5,14,15]. As such, the production of formate over supported Pd nanoparticles will be boosted when bicarbonate is used instead of gaseous $CO_2$ as the C1 source[16]. However, such evolution of $CO_2$ to formate through bicarbonate intermediates actually experiences a thermodynamic sink in the reaction, because the bicarbonate species are considerably more stable than the parent $CO_2$ and final formate. As a result, further hydrogenation of bicarbonate is disfavored and requires the enhanced hydrogenation ability of a heterogeneous catalyst. Therefore, the development of a catalyst that can directly activate $CO_2$ for hydrogenation to formate is highly desirable.

Additionally, for the route through bicarbonate, it has been reported that $CO_2$ can also be chemically captured by N-containing organic bases such as amines[17]. More importantly, such an activation pattern of $CO_2$ is thermally reversible owing to the relatively weak chemical interaction between $CO_2$ and organic bases, which means that the thermodynamic sink caused by bicarbonate in the evolution of $CO_2$ to formate can be avoided if this new pattern of activation can be employed in the process of $CO_2$ conversion[18,19]. In our previous work, we have developed a new type of Schiff-base-modified gold catalyst, which, unlike conventional catalysts, performs well in the decomposition of formic acid to give $H_2$[20]. Taking into account the capability of gold nanocatalysts in various chemo-selective hydrogenation reactions[21-24] as well as the reversible behavior of a catalytic process[16,25], this kind of catalyst offers great potential in the $CO_2$ transformation to formate by facilitating the activation of $CO_2$ and its subsequent hydrogenation with the help of gold nanoparticles.

Here, we propose an unusual $CO_2$ activation pathway that, through a weakly bonded carbamate zwitterion complex on the Schiff-base-mediated gold catalyst, can effectively avoid the thermodynamic sink of the conventional bicarbonate reactant, and lead to the direct hydrogenation of $CO_2$ to formate.

## Results

**Theoretical understanding of $CO_2$ activation.** Using density functional theory (DFT), we first calculated the chemical interactions between $CO_2$ and N-containing organic bases, including the imine Schiff base and various organic amine bases using the conductor-like screening model (COSMO)[26,27]. As shown in Table 1, the $CO_2$ molecule can be directly activated by organic bases in polar water solvent to give a carbamate zwitterion intermediate. The free energies for chemisorption of $CO_2$ ($\Delta G_{298}$) are less negative resulting from the weak chemical interaction between $CO_2$ and N-containing bases. Geometrically, the distance between the $CO_2$ molecule and the Lewis base center in this intermediate is short (1.61–1.67 Å), and this results in an increase in the $C=O$ bond length from 1.16 to 1.22 Å and a distortion of the linear $O=C=O$ bond angle to ~137°. This suggests that although the chemical interactions between $CO_2$ and organic bases are relatively weak, the $CO_2$ molecule can still be well-activated through a non-bicarbonate route in a polar solvent environment.

**Synthesis and characterization of gold nanocatalysts.** Gold nanocatalysts supported on two different alkyl-primary amines-based organic–inorganic hybrid silica materials, namely Au/$SiO_2$-$NH_2$ and Au/$SiO_2$-Schiff, were synthesized according to our previous report using a facile wet-chemistry method[20]. Briefly, Au/$SiO_2$-Schiff was prepared by aldimine condensation of (3-aminopropyl)triethoxysilane (APTES) with formaldehyde, whereas Au/$SiO_2$-$NH_2$ was obtained by cohydrolysis of APTES and tetraethyl orthosilicate (TEOS), followed by a reduction of the gold precursor with $NaBH_4$. Both samples exhibited a metal

---

**Table 1 Structural parameters of $CO_2$ bound to N-containing organic bases as zwitterionic Lewis base (LB) adducts**

| LB | $\Delta G_{298}$ (Kcal mol$^{-1}$) | r(N−$CO_2$) (Å) | r(C = O) (Å) | α(O−C−O) (°) |
|---|---|---|---|---|
| Propylamine | −4.40 | 1.61 | 1.22 | 136 |
| Ethylamine | −4.37 | 1.62 | 1.22 | 137 |
| Diethylamine | −2.19 | 1.66 | 1.22 | 137 |
| Triethylamine | 0.83 | 1.66 | 1.22 | 136 |
| N-propylmethanimine | −1.14 | 1.61 | 1.23 | 136 |

B3LYP/def2-TZVP + COSMO with a solvent environment of water. In a non-polar n-hexane environment, the $\Delta G_{298}$ and changes in the $CO_2$ geometry were negligible

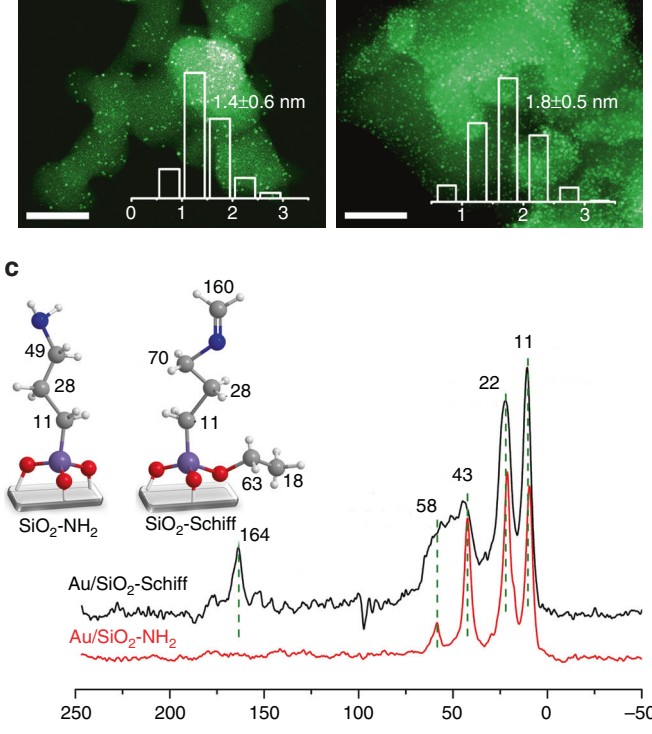

**Fig. 1** Structural characterization of the two different functionalized gold catalysts. HAADF-STEM images of the prepared Au/SiO$_2$-Schiff **a** and Au/SiO$_2$-NH$_2$ **b** catalysts. Scale bar, 50 nm. $^{13}$C CP-MAS NMR spectra of Au/SiO$_2$-Schiff and Au/SiO$_2$-NH$_2$ **c**. The insert shows the results of the quantum mechanics calculation of the NMR shifts

loading of ~1.5 wt%, which was determined by inductively coupled plasma mass spectrometry (Supplementary Table 1).

As indicated by high-angle annular dark-field scanning transmission electron microscopy (HAADF-STEM) (Fig. 1a, b), the different synthetic procedures gave a similar distribution of small gold nanoparticles, with a size of ~1.5 nm (obtained by image analysis of more than 300 particles). By further careful examination with the aberration-corrected HAADF-STEM technique, gold species were found to be dispersed primarily as sub-nanoclusters with sizes of <2 nm (Supplementary Fig. 1). Meanwhile, single-atom gold species were also observed on both the catalysts, but with only a rather low fraction in the entire gold distribution. $^{13}$C CP-MAS NMR spectra suggested that both samples predominantly contained alkylamine groups on the support, but also some unhydrolyzed alkoxysilane groups (Fig. 1c). This was inferred from the chemical shifts at $\delta = 11$, 22, 43, and 58 ppm, as well as the results from quantum mechanics calculations. More interestingly, residual Schiff base (−N = C−) groups were also found in the Au/SiO$_2$-Schiff sample, for which a distinctive resonance peak with a carbon chemical shift at $\delta = 164$ ppm was detected, which was in good agreement with the calculated NMR shift of the imine group (Fig. 1c). The corresponding stretching vibration of this functional group was observed at 1659 cm$^{-1}$ in the FT-IR spectrum (Supplementary Fig. 2b), and was consistent with our calculated value of 1653 cm$^{-1}$. Accordingly, as revealed by X-ray photoelectron spectrometry (XPS), an additional N-containing species appeared in the spectrum of the Au/SiO$_2$-Schiff sample, with the lower-binding energy species belonging to the −N = C− group (Supplementary Fig. 3). Moreover, the gold component of both

catalysts was found to exhibit predominantly weak electronegativity and to possess a low content of positive gold species, which were ascribed to the electron-donor capability of nitrogen groups toward the small gold clusters and the presence of some of the isolated gold cations on the support, respectively (Supplementary Fig. 4). Thus, we successfully obtained two gold nanocatalysts with or without the modification by Schiff base on the support.

**Catalytic hydrogenation of CO$_2$ in liquid phase**. The catalytic performance of the as-prepared gold nanocatalysts for CO$_2$ hydrogenation was studied at 90 °C in a H$_2$/CO$_2$ mixture (80 bar) with triethylamine (NEt$_3$) as an additive in the liquid phase. As shown in Table 2, in contrast to the completely inert behavior of Au/SiO$_2$ (entry 1 in Table 2), the CO$_2$ hydrogenation activity of the gold nanocatalysts was greatly enhanced by the presence of an organic base on the support (entries 2 and 3 in Table 2). Moreover, the Schiff-base-modified gold nanocatalyst was found to be considerably more active than its Au/SiO$_2$-NH$_2$ counterpart. Meanwhile, as expected, such an evolution of CO$_2$ only occurred in polar solvents. For example, in water and methanol, the TON reached 9624 and 9806, respectively (entries 4 and 5 in Table 2), whereas in the non-polar solvents of *n*-hexane (entry 7 in Table 2) and cyclohexane (entry 8 in Table 2), the conversion of CO$_2$ was negligible. The reaction was optimized by experimentally tuning the mixture of methanol and water (Supplementary Table 2), and a TON value as high as 14,470 was achieved in a H$_2$O/methanol mixture containing 20% H$_2$O (vol/vol, entry 2 in Table 2). This TON was comparable with the best result reported in the literature under similar reaction conditions of the heterogeneous conversion of CO$_2$ (Supplementary Table 3).

Bicarbonate is a thermodynamically stable form of CO$_2$ in an alkaline environment, and when bicarbonate is used instead of gaseous CO$_2$ as the precursor for hydrogenation, the production of formate is always increased over Pd-based nanocatalysts[16,28]. However, this is not the case for our Au/SiO$_2$-Schiff catalyst. That is, when bicarbonate was used as the carbon source, the conversion rate decreased inversely, with a TON of only a few hundred (entries 9 and 11 in Table 2). In contrast, when gaseous CO$_2$ was re-introduced into the autoclave, the activity of the catalyst was greatly recovered (entries 10 and 12 in Table 2). This indicated that a non-bicarbonate CO$_2$ hydrogenation route occurred over the Au/SiO$_2$-Schiff nanocatalyst, in which a direct catalytic CO$_2$ hydrogenation was involved[29]. Notably, to date, no heterogeneous catalyst has been reported for the direct catalytic CO$_2$ hydrogenation to formate.

Moreover, another two Schiff-base-modified gold catalysts, where one had a larger metal particle size (~3.2 nm) and the other contained only the single-atom gold species, were also prepared by the surface-functional group promoted in situ reduction method[20] and the milling-mediated solid reduction method, respectively (Supplementary Figs. 5, 6), and were then evaluated at the same reaction conditions. However, the superior activity of the Au/SiO$_2$-Schiff catalyst was greatly reduced or even quenched (entries 13 and 14 in Table 2). This suggests that the hydrogenation of CO$_2$ over gold nanocatalysts exhibits a great size-dependent behavior, with the small sub-nanoclusters being more efficient than the large-sized nanoparticles; however, the single-atom gold species were ineffective as a catalyst in the catalytic hydrogenation of CO$_2$. A similar size dependency of catalytic activity has also been reported previously for the CO oxidation over Au/FeO$_x$ materials[30].

## Discussion

We have successfully achieved the direct catalytic CO$_2$ hydrogenation to formate over amino-functionalized gold

**Table 2 Catalytic performance of the gold nanocatalysts for $CO_2$ hydrogenation**

| Entry | Solvent | $P(H_2)/P(CO_2)$ MPa | Base | Time/h | HCOOH/M | TON[a] |
|-------|---------|----------------------|------|--------|---------|--------|
| 1[b] | $H_2O/CH_3OH$ | 5.0/3.0 | $NEt_3$ | 12 | – | – |
| 2 | $H_2O/CH_3OH$ | 5.0/3.0 | $NEt_3$ | 12 | 0.518 | 14,470 |
| 3[c] | $H_2O/CH_3OH$ | 5.0/3.0 | $NEt_3$ | 12 | 0.08 | 1026 |
| 4[d] | $H_2O$ | 5.0/3.0 | $NEt_3$ | 12 | 0.689 | 9624 |
| 5[d] | Methanol | 5.0/3.0 | $NEt_3$ | 12 | 0.702 | 9806 |
| 6[d] | Ethanol | 5.0/3.0 | $NEt_3$ | 12 | 0.297 | 4148 |
| 7[d] | $n$-Hexane | 5.0/3.0 | $NEt_3$ | 12 | 0.025 | 349 |
| 8[d] | Cyclohexane | 5.0/3.0 | $NEt_3$ | 12 | 0.013 | 181 |
| 9 | $H_2O/CH_3OH$ | 5.0/0 | $KHCO_3$ | 5 | 0.009 | 251 |
| 10 | $H_2O/CH_3OH$ | 5.0/3.0 | $KHCO_3$ | 5 | 0.107 | 2989 |
| 11 | $H_2O/CH_3OH$ | 5.0/0 | $NaHCO_3$ | 5 | 0.007 | 195 |
| 12 | $H_2O/CH_3OH$ | 5.0/3.0 | $NaHCO_3$ | 5 | 0.063 | 1760 |
| 13[e] | $H_2O/CH_3OH$ | 5.0/3.0 | $NEt_3$ | 12 | – | – |
| 14[f] | $H_2O/CH_3OH$ | 5.0/3.0 | $NEt_3$ | 12 | 0.021 | 1207 |

Reaction conditions: 5 mg $Au/SiO_2$-Schiff catalyst, 90 °C, 10 mL reagent, 15 mmol base, 600 rpm, $H_2O/CH_3OH$ (20:80 vol/vol)
[a]The TON is calculated by the Supplementary Equation 1. The number of replicates for each experiment was $n \geq 2$
[b]Catalyst was 5 mg $Au/SiO_2$
[c]Catalyst was 5 mg $Au/SiO_2$-$NH_2$
[d]Catalyst was 10 mg $Au/SiO_2$-Schiff
[e]The single-atom catalyst was prepared by solid-state reduction method
[f]The 3.2-nm nanocatalyst was prepared by surface functional group-promoted in situ reduction method

nanocatalysts. However, the organic base on the support has a great impact on the catalytic activity, with the $Au/SiO_2$-Schiff catalyst being superior to $Au/SiO_2$-$NH_2$. To gain further insight into the unique behavior of $Au/SiO_2$-Schiff and the discrepancy between the organic bases in terms of $CO_2$ hydrogenation, in situ diffuse reflectance infrared Fourier transform spectroscopy (in situ DRIFTS) was employed to monitor the adsorption and activation of $CO_2$ over these two samples. As shown in Fig. 2a, on exposure of $Au/SiO_2$-$NH_2$ to $CO_2$ in the gas phase, a distinct absorption band originating from $CO_2$ was observed, which was similar to a previous report of $CO_2$ adsorption on amine-grafted SBA-15[31]. Accordingly, the peaks at 1488 and 1621 $cm^{-1}$ were ascribed to the deformation vibration of the $NH_3^+$ species[32], and those at 1431 and 1329 $cm^{-1}$ were associated to the skeletal symmetric stretching vibrations of $COO^-$ with its asymmetric mode at 1568 $cm^{-1}$ [33]. These results suggest that an ionic carbamate species forms on the $Au/SiO_2$-$NH_2$ catalyst as a result of proton transfer between two neighboring amine groups. In addition, the low-intensity absorption band at 1690 $cm^{-1}$ was corresponding to the vibration of a carbonyl group derived from the surface-bound carbamate, and no bicarbonate species was detected[31]. As for the $Au/SiO_2$-Schiff catalyst, a $NH_3^+$ deformation vibration and a $COO^-$ skeletal stretching vibration were also detected, as was observed for the $Au/SiO_2$-$NH_2$ catalyst, because of the presence of the alkylamine on the support. In addition, a remarkable absorption peak at 1712 $cm^{-1}$ was observed for the $Au/SiO_2$-Schiff catalyst, which indicated a new adsorption pattern for $CO_2$ resulting from the interaction between the Schiff base and $CO_2$. More importantly, as seen in Fig. 2b, on evacuation of the vessel containing the $CO_2$-saturated $Au/SiO_2$-Schiff catalyst, the intensity of this new band decreased considerably faster than that of the ionic carbamate species. This suggested that the new species were less stable, and could be removed more easily than the ionic carbamate species. Considering that the formation of the ionic carbamate resulted from the transfer of a proton between two neighboring amines, only primary and secondary amines can capture $CO_2$ as an ionic carbamate[31,34]. In contrast, the lack of a proton in the Schiff base (-C = N-) prevents it from acting as a proton donor. It can thus act only as a Lewis base for $CO_2$ adsorption, that is, affording a zwitterion adduct on the support[35–37]. Indeed, we have also simulated the chemisorption of $CO_2$ on the model of a $Au_{55}$ sub-nanocluster (~1.2 nm)

accommodated with an alkyl-imine Schiff base. As shown in Supplementary Fig. 7, gaseous $CO_2$ was found to be captured at the gold/Schiff base interface through a zwitterion intermediate, with an adsorption energy of 0.2–0.4 eV and its oxygen atoms bonded on the low-coordinated sites of the gold nanocluster. The C-O stretch vibrational frequency was calculated to be in the range of 1701–1767 $cm^{-1}$ for various possible adsorption patterns, which was in agreement with the spectral observations.

On the other hand, to elucidate the origins of the variation in the $Au/SiO_2$-$NH_2$ catalyst, we compared the thermodynamic stability of the ionic carbamate and carbamate zwitterion species derived from the primary amine groups (Supplementary Fig. 8). The ionic carbamate was found to be thermodynamically more stable than the carbamate zwitterion, with a free energy for the chemisorption of $CO_2$ ($\Delta G_{298}$) of approximately −8.89 kcal $mol^{-1}$ in water. Correspondingly, this also led to a decrease of the distance between the Lewis base center and $CO_2$ (~1.40 Å). As a result, this may also lead to a thermodynamic sink in $CO_2$ evolution like that occurring through the bicarbonate intermediate path. In contrast, the relatively poor thermal stability of the zwitterion adduct probably offers an important chance to serve as a highly active intermediate for $CO_2$ transformation[36]. Such an assumption has further been supported by the in situ DRIFTS experiments of $CO_2$ hydrogenation. As shown in Fig. 2c, with only the surface-bonded ionic carbamate on the $Au/SiO_2$-Schiff catalyst by decompression of $CO_2$ in the gas phase, no characterized peak of the formate product was detected under the operational pressure of $H_2$ (10 bar). In sharp contrast, with the presence of the carbamate zwitterion on the catalyst surface, a new absorption peak at 1590 $cm^{-1}$, which arose from the stretching vibration of the formate species[38], was appreciably observed with an increase of the exposure time (Fig. 2d).

In addition, because the evolution of $CO_2$ to formate in our experiments requires the transfer of reactants (both $CO_2$ and $H_2$) from the gas to the liquid phase and then finally to the catalyst surface, the reaction kinetics insights are also of great significance for understanding the catalytic mechanism of $CO_2$ hydrogenation. We therefore further studied the dependence of the hydrogenation rate on the reaction conditions for the $Au/SiO_2$-Schiff catalyst. As shown in Supplementary Table 4, the rate of $CO_2$ hydrogenation was strongly dependent on the reaction temperature. The calculated turnover frequency value increased

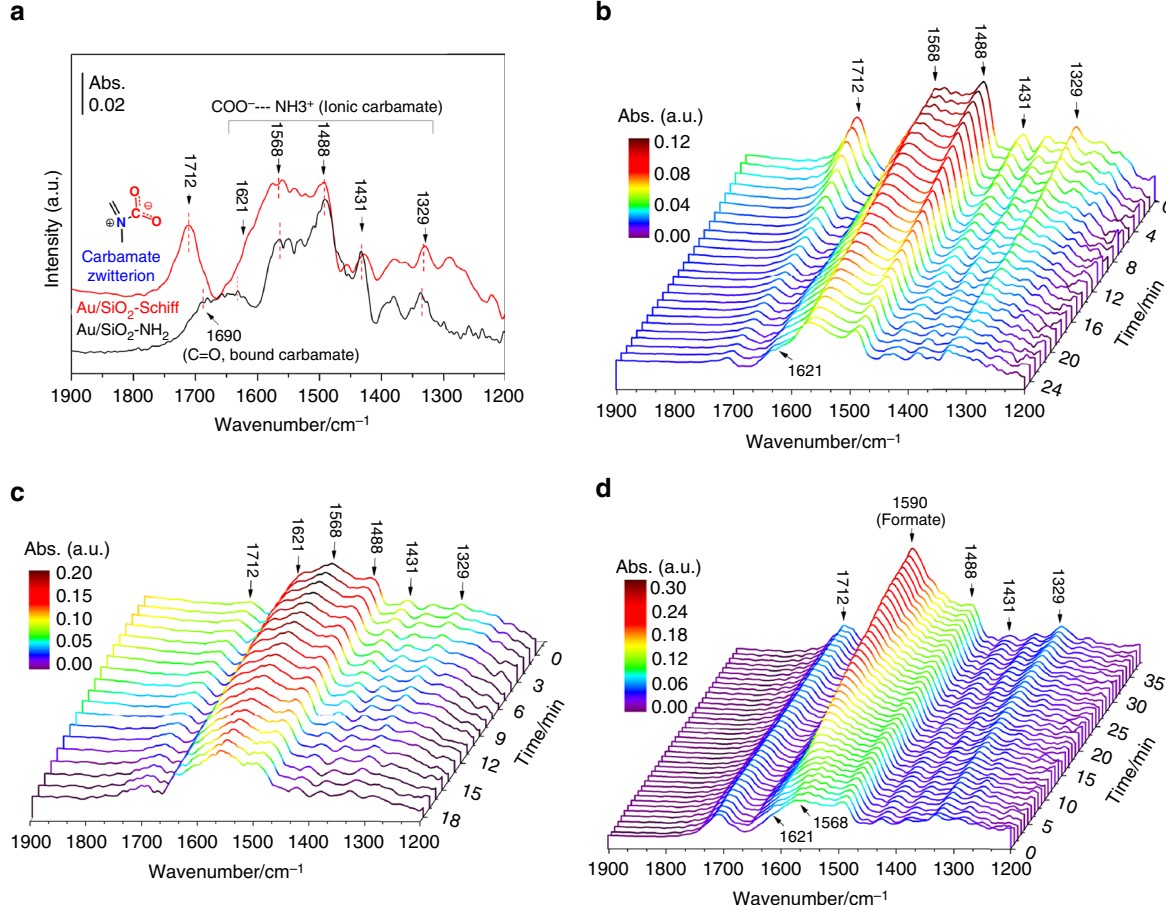

**Fig. 2** Spectral analysis of the gold catalysts during $CO_2$ adsorption/hydrogenation. **a** FTIR spectra of Au/SiO$_2$-NH$_2$ and Au/SiO$_2$-Schiff under a humid $CO_2$ atmosphere. **b** Time-resolved DRIFTS spectra of the Au/SiO$_2$-Schiff catalyst after wet-$CO_2$ evacuation. In situ DRIFTS spectra of the $CO_2$ hydrogenation reaction over Au/SiO$_2$-Schiff catalyst without **c** and with **d** the presence of a surface-bonded carbamate zwitterion

from 195 to 1950 h$^{-1}$ as the temperature was increased from 60 to 90 °C. The apparent activation energy ($E_a$) estimated from the Arrhenius plot was ~76 kJ mol$^{-1}$ (Supplementary Fig. 9), which differed greatly from the previously reported value for the bicarbonate hydrogenation route[24]. Figure 3 shows the turnover rates for formate production over the Au/SiO$_2$-Schiff catalyst as a function of the H$_2$ pressure [$P$(H$_2$)] and $CO_2$ pressure [$P$($CO_2$)] in the autoclave. The formation rate was linearly dependent on the pressure of H$_2$, although there were two different slopes in the high- and low-pressure ranges. This behavior showed that the diffusion and activation of H$_2$ on a gold surface may be the bottleneck for $CO_2$ hydrogenation at lower $P$(H$_2$), whereas the elementary step concerned with the split-H attack becomes the kinetic-controlling step when $P$(H$_2$) is above 30 bar[24]. In contrast, the rate of $CO_2$ conversion was independent of the $CO_2$ pressure. This was because the polar solvent with added NEt$_3$ in our reaction served as a $CO_2$ reservoir, in which gaseous $CO_2$ was captured as carbamate intermediates. Different organic base additives in liquid may therefore exhibit different effects on the hydrogenation activity of the catalyst. As shown in Supplementary Table 5, NEt$_3$ showed a much higher conversion than the other organic amines studied. Interestingly, N-containing organic bases without a hydrogen on the Lewis base centers, such as NEt$_3$ and 1,8-diazabicyclo[5.4.0]undec-7-ene (DBU), showed considerably higher $CO_2$ conversions compared with those containing a proton, like primary or secondary amines. This was because Lewis bases such as NEt$_3$ and DBU can capture $CO_2$ through a weakly bonded carbamate zwitterion complex in a

polar solvent environment, whereas primary or secondary amines will protonate the carbamate zwitterion to yield an ionic carbamate. As described above, the ionic carbamate was less favorable than the carbamate zwitterion for the transfer of $CO_2$ to the supported Schiff base for further hydrogenation. In addition, the durability of the Au/SiO$_2$-Schiff catalyst was also tested. As shown in Supplementary Table 6, a gradual decrease of the catalytic activity occurred, with a TON value of 5922 at the third catalytic cycle, which was caused by the coalescence of small-sized gold to large particles (Supplementary Fig. 10).

Finally, we performed a mechanistic study of $CO_2$ hydrogenation with DFT calculations based on our previous model of the Au/SiO$_2$-Schiff catalyst under the COSMO solvent effect of water. As shown in Fig. 4, the dissociation of H$_2$ to activated H species occurred on the low-coordinated corner sites of gold nanoclusters, with a barrier of 0.67 eV (TS-1) and a weak exothermic contribution (~0.10 eV). As mentioned above, the $CO_2$ molecule can be facilely captured as a zwitterion intermediate on the interface of Schiff base and gold nanocluster, which can then be hydrogenated by the activated H to an HCO$_2$ intermediate on the gold surface, with a barrier of 1.00 eV (TS-2) and acts as the rate-determining step in the whole of the elementary steps. With the assistance from another H adatom, the HCO$_2$ can easily be further hydrogenated to the *cis*-HCOOH on the catalyst surface by passing a barrier of 0.58 eV (TS-3), which subsequently desorbs from the gold surface to obtain the desired *trans*-HCOOH product.

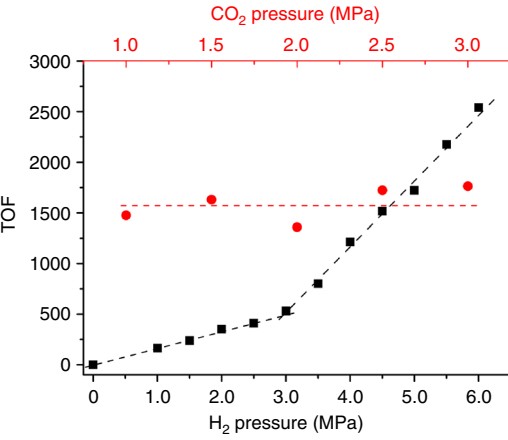

**Fig. 3** H$_2$ (black square) and CO$_2$ (red circle) pressure-dependent initial reaction rates. Reaction conditions: 5 mg Au/SiO$_2$-Schiff catalyst, 10 mL reagent, 10 mmol NEt$_3$, 600 rpm, H$_2$O/CH$_3$OH (20:80 vol/vol). The solution was pressurized at room temperature with CO$_2$ and completed with H$_2$ to the desired pressure. The system was heated at 90 °C and stirred until a desired formate concentration was reached (0.5–3 h). The initial TOF is calculated by the Supplementary Equation 2

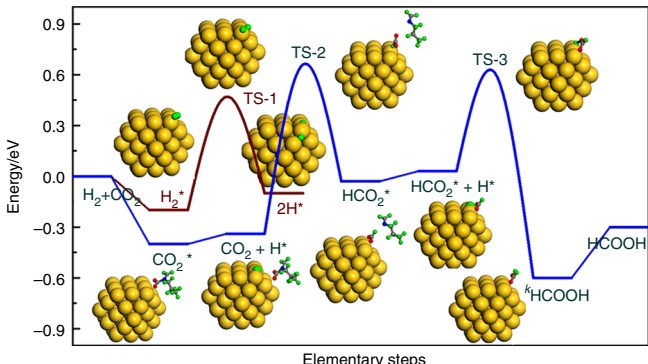

**Fig. 4** Free energy diagram for CO$_2$ hydrogenation over the Au/SiO$_2$-Schiff catalyst. The geometry with a close-packed Au$_{55}$ sub-nanocluster (~1.2 nm) accommodated with an alkyl-imine Schiff base under the COSMO solvent effect of water was used to model the real Au/SiO$_2$-Schiff catalyst in water. Energy profile was constructed based on the DFT calculation analysis of each elementary step

From the combination of catalytic performance, spectrum characterization, kinetic analysis, and DFT calculations, we have therefore proposed a possible catalytic process for CO$_2$ hydrogenation over Au/SiO$_2$-Schiff with a NEt$_3$ additive. As shown in Fig. 5, a gaseous CO$_2$ molecule was captured by the NEt$_3$ through a carbamate zwitterion intermediate, which also served as a reservoir for CO$_2$ in the liquid phase. This weakly chemisorbed CO$_2$ could migrate and be transferred to the gold–Schiff base interface but retained its carbamate zwitterion nature. At the same time, the low-coordinated sites of the gold nanoclusters also participated in the activation and dissociation of H$_2$ to the activated H species. Then, the carbamate zwitterion intermediates were hydrogenated by the H species at the gold–Schiff base interface, and the final formate was thus obtained after a two-step hydrogenation and acid–base neutralization in an alkaline environment. Notably, the electron-rich gold surface, caused by the electron donation from nitrogen groups, might also be beneficial for the hydrogenation of CO$_2$, since it could offer a more negative hydride and lead to a higher reactivity of the

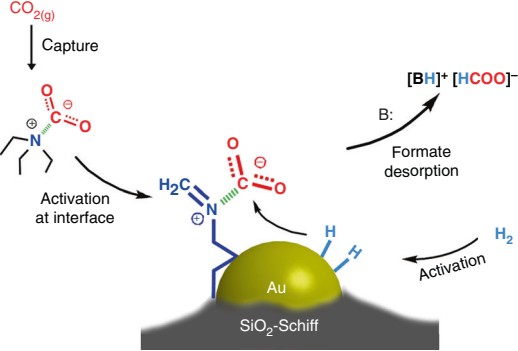

**Fig. 5** Proposed synergetic mechanism for the hydrogenation of CO$_2$ to formate. The activation of CO$_2$ can be achieved through a weakly bonded carbamate zwitterion intermediate at the gold–Schiff base interface, and then hydrogenated by H species to give formate with the help of gold nanoclusters. The electron-rich gold surface is speculated to be beneficial for the hydrogenation of CO$_2$, since it can offer a more negative hydride and lead to a higher reactivity of nucleophilic attack to the carbon center of CO$_2$

nucleophilic attack to the carbon center of CO$_2$[39]. It was therefore also not surprising that the single-atom gold species was inert in the catalytic hydrogenation of CO$_2$ owing to its single site and cation nature. Single-site gold cationic entities have been previously reported as the only active sites in acetylene hydrochlorination[40], which indicates that different gold active sites are involved in the distinguishing catalytic processes.

In conclusion, we have delineated a non-bicarbonate route for the direct catalytic hydrogenation of CO$_2$ to formate using a Schiff-base-modified gold nanocatalyst. Theoretical calculations and in situ high-pressure DRIFTS results demonstrated that the CO$_2$ molecule is activated by the formation of a weak carbamate zwitterionic intermediate at the gold–Schiff base interface, which plays a crucial role on the subsequent transformation to formate. Moreover, the catalytic process benefits from the NEt$_3$ additive in a polar solvent, which allows the formation of a reservoir of CO$_2$ by capturing the gaseous CO$_2$ through the same type of zwitterion intermediate. As a result, the Schiff-base-modified gold nanocatalyst exhibits an unusual catalytic performance compared with those of other functionalized or traditional supported gold nanocatalysts, and shows excellent activity toward CO$_2$ hydrogenation to formate. This finding offers a promising approach for the direct hydrogenation of CO$_2$ and is likely to have considerable implications in the field of CO$_2$ conversion chemistry.

## Methods

**Sample preparation.** *Preparation of Schiff-base-modified SiO$_2$ (SiO$_2$-Schiff)*: The SiO$_2$-Schiff powders were prepared by the APTES with formaldehyde. Typically, 10 mL HCHO solution (37%) was added into 0.12 mol L$^{-1}$ APTES aqueous solutions (500 mL) with stirring at 40 °C for 1 h. The resulting white precipitation was filtered and washed with deionized water, and then transferred into a Teflon-lined autoclave and maintained at 150 °C for 12 h. After the autoclave was cooled down to room temperature naturally, the precipitate was filtered and washed with an excess amount of deionized water. Following drying at 120 °C for 12 h, a stable Schiff-base-functionalized SiO$_2$ support was obtained.

*Synthesis of Au/SiO$_2$-Schiff*: Briefly, 0.5 g SiO$_2$-Schiff was impregnated with an ethanol solution of HAuCl$_4$ (0.095 mM, 400 mL). After stirring at 80 °C for 15 min, 15 mL of NaBH$_4$ (20 mg) aqueous solution was introduced into the mixture to obtain a supported gold catalyst. After stirring for 1 h, the precipitate was filtered and washed with an excess amount of deionized water. Following drying in vacuum at 120 °C for 12 h, the Au/SiO$_2$-Schiff catalyst was obtained.

*Preparation of the Au/SiO$_2$-Schiff catalyst by a surface functional group-promoted in situ reduction method*: Briefly, 0.5 g SiO$_2$-Schiff was impregnated with an aqueous solution of HAuCl$_4$ (0.095 mM, 400 mL). After stirring at 25 °C for 48 h, the mixture was filtered and washed with an excess amount of deionized water. Following drying in vacuum at 120 °C for 12 h, Au/SiO$_2$-Schiff with a diameter of 3.2 nm was obtained.

*Preparation of single-atom gold catalyst by the solid reduction method*: Briefly, 0.2 g SiO$_2$-Schiff was impregnated with an aqueous solution of HAuCl$_4$ (0.005 mM, 200 mL). After stirring at 25 °C for 10 min, the mixture was filtered and washed with ethanol several times. After drying in vacuum at 25 °C for 4 h, the resulting white powders were mixed with 0.1 g NaBH$_4$ and then ground in an agate mortar for 2 h at room temperature. The color change from white to pale yellow indicated the reduction of the gold precursor. After grinding, the resultant product was washed with deionized water and then dried in a vacuum at 50 °C overnight. This solvent-free approach was beneficial for restraining the crystal nucleation and to yield a single-atom catalyst with low metal loadings (~0.1 wt.%).

*Preparation of SiO$_2$-NH$_2$*: The primary amine functionalized silica was synthesized by the cohydrolysis of APTS and TEOS. Typically, 4.0 mL of TEOS was mixed with 4.0 mL of APTES solution and 100 mL of ethanol under stirring at room temperature, to which 10 mL of ammonium hydroxide solution (28 wt%) was added to initiate the hydrolysis of the organosilanes. After stirring at room temperature for 5 h, the precipitate was filtered and washed with an excess amount of deionized water. Following drying at 120 °C for 12 h, the SiO$_2$-NH$_2$ was obtained.

*Preparation of Au/SiO$_2$-NH$_2$*: The Au/SiO$_2$-NH$_2$ was prepared by the similar NaBH$_4$ reduction method as mentioned above, except for the use of a primary amine-modified SiO$_2$ as the support.

Unfunctionalized Au/SiO$_2$ was synthesized by heat treating the Au/SiO$_2$-NH$_2$ in an O$_2$ atmosphere at 800 °C for 20 s. Prior to the heat treatment, Au/SiO$_2$-NH$_2$ was first encapsulated by SiO$_2$ layers through the hydrolysis of TEOS in an alkaline environment. For more details about the synthesis and characterization, please refer to the Supplementary Methods.

**Materials characterization.** The Au loading of the catalysts was measured by inductively coupled plasma atomic emission spectroscopy (ICP-AES) on an IRIS Intrepid II XSP instrument (Thermo Electron Corporation). All the solid-state NMR experiments were performed on a Bruker AvanceIII 600 spectrometer equipped with a 14.1 T wide-bore magnet. The resonance frequencies were 156.4 MHz for $^{13}$C.$^1$H → $^{13}$C CP/MAS NMR experiments were performed on a 4 mm MAS probe with a spinning rate of 12 kHz.$^1$H → $^{13}$C CP/MAS NMR experiments were carried out with a contact time of 5 ms and a recycle delay of 2 s. The chemical shifts were referenced to adamantane with the upfield methine peak at 29.5 ppm. HAADF-STEM images were obtained using a JEOL JEM-2100F at 200 kV. The samples for electron microscopy were prepared by grinding and subsequent dispersion of the powder in ethanol and applying a drop of the very dilute suspension on the carbon-coated grids. XPS was performed on a Kratos Axis Ultra DLA X-ray photoelectron spectrometer equipped with an Al Kα radiation source (1486.6 eV, 15 kV). All binding energies were calibrated with the C1s peak at 284.8 eV for the adventitious carbon. The FTIR spectra were acquired with a spectrometer (BRUKER Equinox 55) equipped with a DLATGS detector and operated at a resolution of 4 cm$^{-1}$.

**In situ DRIFTS experiments.** Spectral analysis of the catalysts during CO$_2$ adsorption/hydrogenation was carried out using an Equinox 55 infrared spectrometer (Bruker) equipped with a high-temperature/high-pressure DRIFTS reactor cell and liquid nitrogen-cooled DLATGS detector.

All spectra were obtained with a resolution of 4 cm$^{-1}$ and an accumulation of 32 scans. The sample cup of the cell was filled with a finely powdered sample. Prior to the CO$_2$ adsorption studies, catalyst samples were pre-treated in situ in a pure He stream (30 mL min$^{-1}$), heated up to 120 °C at a rate of 10 °C min$^{-1}$, then kept at the final temperature for 2 h. After pre-treatment, the cell was cooled to the required temperature in He.

Spectra of CO$_2$ absorption were recorded at 80 °C in a pure CO$_2$ stream (10 mL min$^{-1}$). After adsorption saturation, the system was purged with He (30 mL min$^{-1}$) to obtain the desorption spectra. The CO$_2$ absorption/desorption spectra in a humid atmosphere were realized by bubbling through water.

For the high-pressure DRIFTS experiments, the free wet-CO$_2$ was carefully released from the vessel until the carbamate zwitterion was decomposed completely from the CO$_2$-saturated Au/SiO$_2$-Schiff catalyst, with only surface-bonded ionic carbamate left behind on the catalyst surface (at this moment, CO$_2$ remained in the gas phase with <1 bar). Then, the vessel was sealed and pressured with 10 bar of H$_2$ to trace the evolution of the surface species on the surface. For comparison, to understand the contribution of the carbamate zwitterion species on CO$_2$ conversion, the vessel was directly pressurized with 10 bar of H$_2$ to monitor the spectral evolution of CO$_2$-satuated Au/SiO$_2$-Schiff (with 1 bar of CO$_2$).

**Catalytic activity measurement of CO$_2$ reduction.** Preliminary studies with different catalysts were carried out in a magnetically driven Parr autoclave (50 mL) containing a mixture of H$_2$O-MeOH (20/80 v/v, 10 mL) and base (15 mmol). The solutions were purged three times with high purity CO$_2$, and then pressurized up to 30 bar of CO$_2$ and completed up to 80 bar with H$_2$ ($P$(H$_2$)/$P$(CO$_2$) = 5/3). The system was heated to 90 °C and stirring was started until the reaction was completed. The reaction mixture was transferred into a centrifuge tube and the solid catalyst was separated by centrifugation. Formate product concentrations were monitored by high-performance liquid chromatography (HPLC, Agilent 1100) on

an anion-exclusion column (AminexHPX-87H) using an aqueous H$_2$SO$_4$ solution (5 mM) as the eluent and an ultraviolet detector ($\lambda$ = 210 nm).

For kinetic measurements, 5 mg of Au/SiO$_2$-Schiff was added into the Parr autoclave containing a mixture of H$_2$O-MeOH (20/80 v/v, 10 mL) and base (10 mmol). The solutions were pressurized up to 10–30 bar with CO$_2$ and then completed to 10–60 bar with H$_2$. The system was heated to the required temperature (60–90 °C) and stirring was started (600 rpm). After reaction for the desired time, the reactor was cooled down and depressurized. The reaction mixture was transferred into a centrifuge tube and the solid catalyst was separated by centrifugation. The formate product concentrations were monitored by the method mentioned above.

**Computational methods.** *The quantum mechanics calculations*: The calculation of the NMR shift was carried out using the Gaussian09 quantum chemical package. The geometries were optimized using M062X functional[41], and Ahlrichs's basis sets were used for all atoms at the def2-TZVP level[42]. The second order Møller–Plesset perturbation method[43] and the pcS-2 basis set[44] were used for computing the chemical shift, with the first peak at $\delta$ = 11 from the experiment used as a reference.

The chemical interactions between CO$_2$ and N-containing organic bases were modeled with NWChem software at the B3LYP/def2-tzvp level[45]. Different solvent environments were simulated by the COSMO, with the dielectric constant of *n*-hexane, methanol, and water of 1.89, 32.63, and 78.54, respectively.

The simulation of the chemisorption of CO$_2$ was performed with the program package DMol$^3$ in the Materials Studio of Accelrys Inc[46,47]. Considering the flexibility of the carbon chain, the geometry with a close-packed Au$_{55}$ nanocluster (~1.2 nm) accommodated by an alkyl-imine Schiff base in COSMO of water was used to model the practical Au/SiO$_2$-Schiff catalyst in water. The localized double-numerical basis sets with polarization functions (DNP) were used, and the exchange-correlation functional of PBE was employed. The vibrational frequency was corrected by the experimental measurement of gas CO$_2$ at 2340 cm$^{-1}$. Transition states were obtained by the linear synchronous transit (LST) and quadratic synchronous transit (QST) methods, and were confirmed by the only one imaginary frequency as well as the nudged elastic band (NEB) method.

**Data availability**. The data that support the findings of this study are available from the corresponding author upon request.

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

## Acknowledgements

We acknowledge the National Natural Science Foundation of China (21476226, 21676045, 21776269, 21176037), China Ministry of Science and Technology under contact of 2016YFB0600902, the Strategic Priority Research Program of the Chinese Academy of Sciences (Grant XDA09030101, XDB17020400), Dalian Science Foundation for Distinguished Young Scholars (2016RJ04), the Youth Innovation Promotion Association CAS, the Natural Science Foundation of Liaoning Province (201602169), and the Fundamental Research Funds for the Central Universities (DUT15LK29) for financial support. The calculations were performed at Shanghai Supercomputing Center.

## Author contributions

Q.L. designed the project and performed the catalyst preparation, characterizations, and catalytic tests. X.Y. performed the DFT calculations and completed the paper. L.L. assisted with the DRIFTS characterization. S.M. assisted with the aberration-corrected HAADF-STEM characterization. Y.L. collaborated with the quantum mechanics calculation of the NMR shifts. Yq.L. and T.Z. participated in beneficial discussions and helped to prepare the manuscript. X.Y., X.W., and Y.H. proposed, planned, designed, and supervised the project. All authors reviewed and commented on the manuscript.

## Additional information

**Competing interests:** The authors declare no competing financial interests.

