## [Peer Review File · Nature Communications]

Reviewer #1 (Remarks to the Author):

The manuscript describes the catalytic activity of a Schiff-base-mediated gold nanoparticle catalyst in the hydrogenation of CO₂ to formate, an extremely important and timely topic. This manuscript is in general clearly written and the results are definitely of interest to the readers. However, this work requires some significant improvement before publication is considered.

1. In addition to references 1 and 2, recent reviews and prospective should be cited, e.g. Chem. Soc. Rev. 2016, 45, 3954; ACS Energy Lett. 2017, 2, 188; etc.
2. Theoretical understanding of CO₂ activation is unnecessary as these results are commonly known and no new insights were provided. The role of the "zwitterionic Lewis base" in this CO₂ hydrogenation is unclear. There is little evidence in the experimental results to confirm the authors' proposed mechanism. The theoretical work should model the catalytic process to examine the proposed effects.
3. Similar materials have been reported by the same group for the selective dehydrogenation of the formic acid (Energy Environ. Sci., 2015, 8, 3204), and thus it is conceivable that they can also catalyze the reverse reaction. The authors should clearly demonstrate what is new, with experimental or/and DFT support.
4. How were TOF and TON determined? Are the catalysts reusable? How are their activities compared to the homogeneous counterparts? For heterogeneous systems, it is very important to show their advantages.
5. How was the RDS determined, "dissociation of H₂ to activate H species"?

Several typos should be corrected:

1. title: "catalysis" should be "catalyst".
2. Line 41, " and that this" should be "which".
3. Line 50, remove "of course".

Reviewer #2 (Remarks to the Author):

This paper reports a study of solution phase CO₂ reduction at gold nanoparticles supported on organic-base-modified silica. The principle point of interest is that reasonable activity and turn over numbers are observed for conversion of CO₂ directly to formate, avoiding bicarbonate. The authors have done a reasonable number of control experiments, and the basic finding of good activity for direct CO₂ reduction, appears to be justified adequately. I do not feel expert enough in this particular type of chemistry to assess some of the details of the proposed mechanism, but I feel that the results should be of interest to many in the CO₂ reduction area. Therefore, I recommend publication.

The level of detail given about the materials synthesis, experimental procedures, and data analysis is lower than I would like to see. For example, XPS is presented, showing significant shifts in Au binding energies for the base-modified support, compared to plain silica, but without any discussion. Given that binding energies are sensitive to both particle size and chemical environment, it would be interesting to know what the shifts tell us about the catalytic activity. I would also be interested in knowing more about the S/TEM characterization. For example, what is the size distribution, and does it include both nanoparticles and small clusters/isolated atoms? Previous S/TEM characterization of supported gold catalyst (e.g. Hutching et al.) suggested that the active species in some cases are not the nanoparticles, but rather smaller clusters on the support.

Reviewer #3 (Remarks to the Author):

The authors have identified a Schiff base mediated gold nanocatalyst for the efficient hydrogenation of carbon dioxide to formate. The TON is high, 14470 over 12h at 90 °C. at ~ 80 Bar pressure of hydrogen and CO₂. The best results, according to authors, are based on the reduction of the CO₂-Au silica Schiff complex in water over true bicarbonate like species. The authors have also characterized the complex by IR. The authors also surmise that the lack of proton on the Schiff base nitrogen leads to better reactivity of CO₂ complex. However, in 20 % water such analysis may have problems.

Based on catalytic activity, spectral analysis (IR) and DFT calculation, the authors have proposed a mechanism for CO₂ reduction to formate (Figure 3). However, looking at a published review in 2016 (cited as ref. 12), the efficacy for CO₂ reduction to formate, the present work's TOF of 1206 h⁻¹ is the highest among studied Au catalysts. There are other catalysts, which give higher TOF h⁻¹ in the 80 -120 °C range (Table 1 of the review). This referee feels that the work is interesting and publishable in a specialized journal and not in Nature Communications

Dear Prof. Bottari and Reviewers,

On behalf of my co-authors, we sincerely thank you and all reviewers for their valuable and constructive comments that we have used to improve the quality of our manuscript (ID: NCOMMS-17-00690).

According to the referees' comments, a point-by-point response to the referees' questions was provided as shown in the following context. Our changes made to the manuscript are highlighted in red color. We hope that this revised manuscript will be acceptable for publication in your journal. Thank you very much for your consideration.

With best wishes,

Yanqiang Huang,
Dalian Institute of Chemical Physics,
457# Zhongshan Road, Dalian, 116023, China
Tel: +86-411-84379416
Fax: +86-411-84685940
E-mail: yqhuang@dicp.ac.cn

Responses to the Referees' comments

Referee: 1

Comments:

The manuscript describes the catalytic activity of a Schiff-base-mediated gold nanoparticle catalyst in the hydrogenation of CO₂ to formate, an extremely important and timely topic. This manuscript is in general clearly written and the results are definitely of interest to the readers. However, this work requires some significant improvement before publication is considered.

1. In addition to references 1 and 2, recent reviews and prospective should be cited, e.g. Chem. Soc. Rev. 2016, 45, 3954; ACS Energy Lett. 2017, 2, 188; etc

Response: As suggested, we have expanded our list of references to the recent publications in this field.

2. Theoretical understanding of CO₂ activation is unnecessary as these results are commonly known and no new insights were provided. The role of the "zwitterionic Lewis base" in this CO₂ hydrogenation is unclear. There is little evidence in the experimental results to confirm the authors' proposed mechanism. The theoretical work should model the catalytic process to examine the proposed effects.

Response: Thank you for your insight. Despite the fact that ionic carbamate as an activated specie has been involved in several strategies for CO₂ hydrogenation, zwitterionic adduct derived from the reaction of CO₂ with hydrogen-lacking nitrogen center was rarely captured and its role in promoting the catalytic CO₂ transformation was reported for the first time. Thus, theoretical understanding the formation of carbamate zwitterion complex is necessary as it plays an important role for CO₂ activation and subsequent transformation in this study (see below). We shortened this part in the revised manuscript as suggested by the referee (page 4).

As suggested, we have further simulated the chemisorption of CO₂ on the model of

Au/SiO₂-Schiff catalyst with an Au₅₅ nanocluster (~1.2 nm) and a silylation imine Schiff base on it. As seen in Supplementary Fig. 5, the CO₂ was found to be captured at the gold-Schiff base interface through a zwitterion intermediate, with one of its oxygen bonded on the low-coordinated corner site of gold nanocluster, and the C-O stretch vibrational frequency was calculated to be 1769 cm⁻¹, which was in good agreement with the spectra observations. Experimentally, *in situ* high-pressure DRIFTS were used to identify the role of carbamate zwitterion and confirmed the proposed mechanism. However, as shown in Figure 1a and 1b, due to the relatively weak chemisorption of CO₂ over Schiff base, the intensity of the carbamate zwitterion was sensitive to the CO₂-pressure and diminished completely under sub-atmospheric pressure, thus it is difficult to directly detect the evolution of carbamate zwitterion at H₂ atmosphere without CO₂ in presence. Experimentally, by rigidly controlling the CO₂-pressure in gas phase, as seen in Figure 1c, when with the only surface-bonded ionic carbamate on Au/SiO₂-Schiff catalyst, no characterized peak of formate product was detected under the operational pressure of H₂ (10 bar). It indicated that the surface-bonded ionic carbamate is inactive for further hydrogenation at the given conditions. In sharply contrast, with the presence of carbamate zwitterion on catalyst surface at the same H₂ pressure, a new absorption peak corresponding to formate was observed at ca. 1590 cm⁻¹ with a rise in the exposure time (Figure 1d). Collectively, it indicated that the weakly bonded carbamate zwitterion plays a crucial role on not only the activation of CO₂, but also the subsequent transformation into formate. The corresponding discussion was added in the revised manuscript (page 8-9).

Figure 1. Time-resolved DRIFTS spectra of the whole spectra (a) and narrow spectra (b) for Au/SiO₂-Schiff catalyst after CO₂ evacuation under humid conditions. *In situ* high-pressure DRIFTS spectra of CO₂ hydrogenation reaction over Au/SiO₂-Schiff catalyst without (c) and with (d) the presence of surface-bonded carbamate zwitterion. (the CO₂-pressure (less than 1 bar) was rigidly controlled to realize the absence/presence of surface-bonded carbamate zwitterion, then the vessel was sealed and pressured with 10 bar of H₂ to track the reaction profile)

3. Similar materials have been reported by the same group for the selective dehydrogenation of the formic acid (*Energy Environ. Sci.*, 2015, 8, 3204), and thus it is conceivable that they can also catalyze the reverse reaction. The authors should clearly demonstrate what is new, with experimental or/and DFT support.

Response: In our previous report, the similar material was synthesized and tested in the selective dehydrogenation of formic acid, which exhibited excellent catalytic activity. It thus inspires us to conduct further research on the reverse reaction of the CO₂ hydrogenation to formate. Indeed, it was found that such Schiff-base-mediated gold catalyst shows a remarkable activity in CO₂ hydrogenation to formate, with a turnover number (TON) of up to 14,470 over 12 h at 90 °C, which is comparable to

the best result reported in the literature under similar reaction conditions, and is the first time for the direct CO₂ catalytic hydrogenation to formate over a heterogeneous catalyst. More important, with the combination of theoretical DFT calculations and *in situ* high-pressure DRIFTS, an unusual activation pathway through a new weakly bonded carbamate zwitterion complex was identified on the interface of gold and Schiff base, which can effectively avoid the thermodynamic sink of the conventional bicarbonate reactant or the ionic carbamate on amine, and leads to a direct hydrogenation of CO₂ to formate. We believe that this finding is much important and will open up a new avenue for CO₂ activation, and is likely to have considerable implications in the field of CO₂ transformation and gold catalytic chemistry.

4. How were TOF and TON determined? Are the catalysts reusable? How are their activities compared to the homogeneous counterparts? For heterogeneous systems, it is very important to show their advantages.

Response: After ensuring that the reaction is free of diffusion limitations (the selected stirrer speed is as high as 600 r/min and the catalyst was grinded before use), the TON and initial TOF reported here is calculated according to the equation as follows:

$$TON = \frac{C_{HCOOH} \times V}{n_{Au}} \quad TOF = \frac{C_{HCOOH} \times V}{n_{Au} t}$$

Where C_{HCOOH} is the formate concentration monitored by high performance liquid chromatography (HPLC), n_{Au} is the total mole number of Au atoms in catalyst, V is the volume of reactants, and t is the initial time of the catalytic reaction (where C_{HCOOH} is less than 0.05 M compared with the 1.0 M base added). The detailed calculation methods for TOF and TON have been added in the supporting information.

Experimentally, the durability of the Au/SiO₂-Schiff catalyst was tested at 90 °C in a H₂/CO₂ mixture (80 bar). As seen in Table 1, a gradual decrease of catalytic activity was observed, with a TON value of 5922 in the third catalytic cycle, which was caused by the coalescence of small sized gold to large particles (Supplementary

Fig. 8). As suggested, the recyclability of the catalyst was discussed in the manuscript (Page 10).

Table 1. Reusage of the Au/ SiO₂-Schiff catalyst for the hydrogenation of CO₂ to formic acid.

Cycle	HCOOH /M	TON
1	0.908	12684
2	0.652	9107
3	0.424	5922

Reaction conditions: 10 mg catalyst, 90 °C, 10 mL reagent, 15 mmol base, 12 h, 600 rpm.

Even though the currently reported catalytic efficiency for CO₂ hydrogenation to formate has been greatly promoted, which was the highest among studied gold catalysts and was comparable to the best result reported in the literature under similar reaction conditions of other heterogeneous catalysts, it is still much lower than those of homogeneous catalyst systems (Supplementary Table 3). This is mainly caused by the different catalytic mechanism of these two types of catalytic processes. However, due to the advantages of heterogeneous catalysis in the separation and reuse of materials, the heterogenization of such process is still the main goal in the future. To further improve the durability and activity of our gold nanocatalyst in CO₂ conversion, new strategies for developing more sustainable and more efficient catalysts are still in progress in our lab.

5. How was the RDS determined, "dissociation of H₂ to activate H species"?

Response: As indicated by the kinetic measurements (Fig. 3 in the manuscript), the turnover rate for formate production was independent of the CO₂ pressure in gas phase, thus it suggests that CO₂ and its related steps are not involved into the rate-determining step. On the other hand, the linear dependence toward H₂ pressure indicates that the step participated by the dihydrogen molecule, but not the dissociated H species, is the rate-determining step of reactions. In addition, it is well known that the dissociation of H₂ was usually disfavored over gold nanocatalyst (J. Am. Chem. Soc. 2013, 135, 15244–15250). As a result, we speculated that the dissociation of H₂

into activated H species might be the rate-determining step in the CO₂ conversion over our gold catalyst (Page 10).

Several typos should be corrected:

1. Title: "catalysis" should be "catalyst".

Response: Done.

2. Line 41, " and that this" should be "which".

Response: Done.

3. Line 50, remove "of course".

Response: Done.

Referee: 2

Comments:

This paper reports a study of solution phase CO₂ reduction at gold nanoparticles supported on organic-base-modified silica. The principle point of interest is that reasonable activity and turn over numbers are observed for conversion of CO₂ directly to formate, avoiding bicarbonate. The authors have done a reasonable number of control experiments, and the basic finding of good activity for direct CO₂ reduction, appears to be justified adequately. I do not feel expert enough in this particular type of chemistry to assess some of the details of the proposed mechanism, but I feel that the results should be of interest to many in the CO₂ reduction area. Therefore, I recommend publication.

The level of detail given about the materials synthesis, experimental procedures, and data analysis is lower than I would like to see. For example, XPS is presented, showing significant shifts in Au binding energies for the base-modified support, compared to plain silica, but without any discussion. Given that binding energies are sensitive to both particle size and chemical environment, it would be interesting to know what the shifts tell us about the catalytic activity. I would also be interested in knowing more about the S/TEM characterization. For example, what is the size distribution, and does it include both nanoparticles and small clusters/isolated atoms? Previous S/TEM characterization of supported gold catalyst (e.g. Hutching et al.) suggested that the active species in some cases are not the nanoparticles, but rather smaller clusters on the support.

Response: Thanks for your good suggestion. As suggested, the materials synthesis, experimental procedures, and data analysis were detailed in the revised manuscript (page 12-14). Moreover, both the DFT calculations and FTIR experiments were further performed to obtain a more comprehensive understanding of catalytic mechanism. The corresponding discussion can be found in the revised manuscript (page 8-9).

As for the XPS spectra results of gold species (Supplementary Fig. 3), it was found that both the gold nanocatalysts exhibited weak electronegativity, which was ascribed to the electron-donor capability of nitrogen-containing functional groups toward the small sized gold species on these two samples. On the other hand, according to our simulation of CO₂ capture on Au/Schiff-base model catalyst (Au₅₅, ~1.2 nm), both the size of gold and its chemical environment are essential for the activation of CO₂, that is, the chemisorption of CO₂ only occurs on the interface between the low coordinated gold site and Schiff base functional group. Therefore, it demonstrated that the gold species with lower binding energy is one of the main factors governing the high activity of our Au/Schiff-base catalysts.

As shown in Figure 1a and 1b (in the manuscript), the different synthetic procedures gave a similar distribution of small sized gold nanoparticles, with a size of ~1.5 nm. By further carefully examining with the aberration-corrected HAADF-STEM technique (Fig. 2), gold species were found primarily dispersed as sub-nanoclusters with size less than 2 nm [ACS Catal. 2011, 1, 2–6]. Meanwhile, single-atom gold species were also observed on both the catalysts, but with only a rather low fraction in the entire gold distribution. In order to distinguish the intrinsic reactivity of different size gold on CO₂ hydrogenation, a gold catalyst with nanoparticles at a diameter of 3.2 nm was also prepared, and evaluated under the same reaction conditions. The represent STEM images were displayed in Supplementary Fig. 4, with the experimental data shown in Table 2 (in the manuscript). In contrast, the superior activity of Au/SiO₂-Schiff catalyst was quenched (entry 13), illustrating the size-dependent behavior of gold catalyst with the only smaller gold nanoclusters effectively involving into catalytic conversion (Nature communications, 2016, 7, DOI: 10.1038/ncomms12905). The corresponding discussion has been added in the revised manuscript (page 8).

Figure 2. (a) Typical aberration-corrected HAADF-STEM images of Au/SiO₂-Schiff (a,b) and Au/SiO₂-NH₂ (c,d). The Au species are mainly dispersed as nanoclusters (triangle), as well as single atoms (square) and nanoparticles larger than 2 nm (circles).

Referee: 3

Comments:

1. The authors have identified a Schiff base mediated gold nanocatalyst for the efficient hydrogenation of carbon dioxide to formate. The TON is high, 14470 over 12h at 90 °C. at ~ 80 Bar pressure of hydrogen and CO₂. The best results, according to authors, are based on the reduction of the CO₂-Au silica Schiff complex in water over true bicarbonate like species. The authors have also characterized the complex by IR. The authors also surmise that the lack of proton on the Schiff base nitrogen leads to better reactivity of CO₂ complex. However, in 20 % water such analysis may have problems.

Response: Our description of Schiff base as a “proton” lacking in the manuscript might be misleading. In our opinion, the key point for the capture of CO₂ as active carbamate zwitterion intermediate was only feasible with a hydrogen-lacking Lewis-base center like imine, otherwise the formation of the inactive ionic carbamate will be more facile by the chemical reaction of CO₂ with primary or secondary amines. Accordingly, in the revised manuscript, the word of “*non-protonated*” has been replaced by the word of “*hydrogen-lacking*”.

In our report, the most direct evidence for the unusual catalytic performance of Au/SiO₂-Schiff catalyst was revealed by its activity dependence on different carbon sources. As shown in Table 2, when bicarbonate was used as the carbon source, the conversion rate was rather low, with a TON of only a few hundreds (entries 1 and 3). In contrast, when gaseous CO₂ was introduced into the autoclave, the activity of the catalyst was greatly recovered (entries 2 and 4). This indicates that a non-bicarbonate hydrogenation route occurs over the Au/SiO₂-Schiff catalyst. In addition, the gaseous CO₂ hydrogenation via the proposed carbamate zwitterion intermediate was definitely proved by the *in situ* high-pressure DRIFTS (see below).

Table 2. The hydrogenation of bicarbonates in the presence or absence of gaseous CO₂ over the Au/SiO₂-Schiff catalyst.^a

Entry	Solvent	P(H ₂)/P(CO ₂) Mpa	Base	Time /h	HCOOH /M	TON
1	H ₂ O/CH ₃ OH	5.0/0	KHCO ₃	5	0.009	251
2	H ₂ O/CH ₃ OH	5.0/3.0	KHCO ₃	5	0.107	2989
3	H ₂ O/CH ₃ OH	5.0/0	NaHCO ₃	5	0.007	195
4	H ₂ O/CH ₃ OH	5.0/3.0	NaHCO ₃	5	0.063	1760

[a] Reaction conditions: 5 mg Au/SiO₂-Schiff catalyst, 90 °C, 10 mL reagent, 15 mmol base, 600 rpm, H₂O/CH₃OH (20:80 vol/vol).

Experimentally, we also studied the influence of water on the formation of carbamate zwitterion by *in situ* DRIFTS. As shown in figure 3a, on exposure of the Au/SiO₂-Schiff sample to a moist CO₂, the absorption band of the ionic carbamate was kept unchanged, however, the band contributed to the carbamate zwitterion has a clearly shift to lower frequency (from 1719 cm⁻¹ to 1712 cm⁻¹). This red shift can be ascribed to the H-bond effect between the fixed H₂O molecule and O-atom of the carbonyl (J. Phys. Chem. C 2011, 115, 11540–11549).

Figure 3. FTIR spectra of Au/SiO₂-Schiff under a dry/humid CO₂ atmosphere.

In addition, we have also compared the hydrogenation reactivity of these two different surface-bonded CO₂-derived complexes (i.e., ionic carbamate and carbamate zwitterion). However, as shown in Figure 4a and 4b, due to the relatively weak chemisorption of CO₂ over Schiff base, the intensity of the carbamate zwitterion was sensitive to the CO₂-pressure and diminished completely under sub-atmospheric

pressure, thus it is difficult to directly detect the evolution of carbamate zwitterion at H₂ atmosphere without CO₂ in presence. Experimentally, by rigidly controlling the CO₂-pressure in gas phase, as seen in Figure 4c, when with the only surface-bonded ionic carbamate on Au/SiO₂-Schiff catalyst, no characterized peak of formate product was detected under the operational pressure of H₂ (10 bar). It indicated that the surface-bonded ionic carbamate is inactive for further hydrogenation at the given conditions. In sharply contrast, with the presence of carbamate zwitterion on catalyst surface at the same H₂ pressure, a new absorption peak corresponding to formate was observed at ca. 1590 cm⁻¹ with a rise in the exposure time (Figure 4d). These experiments indicate that the weakly bonded carbamate zwitterion is more reactive than the ionic carbamate species, and is crucial for CO₂ activation and transformation, which is consistent with our proposed mechanism.

Based on the discussion above, the presence of water did not alter the formation of carbamate zwitterion as well as the reaction mechanism. With this regard, the spectra of CO₂ absorption/desorption were re-recorded under humid conditions (Figure 2a and 2b in the manuscript) and the *in situ* high-pressure DRIFTS results were added to clarify the proposed mechanism (page 9). Thanks for your insight.

Figure 4. Time-resolved DRIFTS spectra of the whole spectra (a) and narrow spectra (b) for Au/SiO₂-Schiff catalyst after CO₂ evacuation under humid conditions. *In situ* DRIFTS spectra of CO₂ hydrogenation reaction over Au/SiO₂-Schiff catalyst without (c) and with (d) the presence of surface-bonded carbamate zwitterion.

2. Based on catalytic activity, spectral analysis (IR) and DFT calculation, the authors have proposed a mechanism for CO₂ reduction to formate (Figure 3). However, looking at a published review in 2016 (cited as ref. 12), the efficacy for CO₂ reduction to formate, the present work's TOF of 1206 h⁻¹ is the highest among studied Au catalysts. There are other catalysts, which give higher TOF h⁻¹ in the 80-120 °C range (Table 1 of the review). This referee feels that the work is interesting and publishable in a specialized journal and not in Nature Communications.

Response: We agree with referee that the present work's TOF is the highest among the studied gold catalysts, but there are many other catalysts, especially homogeneous catalysts, which give comparable or much higher TOF than ours at similar reaction conditions. However, there are many features of our research on the

understanding the CO₂ activation. First, heterogeneous catalytic transformation of CO₂ is generally realized through a bicarbonate hydrogenation in an alkaline environment, while it suffers from a thermodynamic sink due to the considerable thermodynamic stability of bicarbonate intermediate. While in our research, a route for direct catalytic conversion of CO₂ was first achieved even over a conventional inert gold nanocatalyst, which is comparable to the fastest known heterogeneous catalysts, with a turnover number (TON) of up to 14,470 over 12 h at 90 °C. Second, theoretical calculations and spectra analysis results unambiguously demonstrated that the activation of CO₂ can be achieved through a weakly bonded carbamate zwitterion intermediate derived from simple Lewis base adduct of CO₂ at the gold-support interface, while it can only occur with hydrogen-lacking Lewis-base center in a polar solvent. This finding opens up a new avenue for the direct activation of CO₂, and is likely to have considerable implications in the field of CO₂ conversion and gold catalytic chemistry. Thus, we hope that this revised manuscript will be acceptable for publication in the top journal of *Nature Communications*.

Reviewer #1 (Remarks to the Author):

Most of the concerns have been addressed by the authors and the manuscript was significantly improved. Publication is recommended once the last issue is resolved.

The most important point the authors claim appears to be the new CO₂ reduction pathway based on the different reactivities of the Au/SiO₂-NH₂ and Au/SiO₂-Schiff materials. Since they have done some preliminary DFT studies, could they further show that the Schiff-base activated CO₂ is more reactive towards reduction (i.e. with a lower activation energy barrier)?

Reviewer #2 (Remarks to the Author):

The authors partially addressed my comments about wanting more experiment detail, and discussion of XPS and STEM, but in a very cursory fashion. The XPS shifts presumably tell us something about the state of the catalyst, but they only describe this in terms of the "electronegativity" of the gold. Similarly, the STEM results show that atoms, clusters, and nanoparticles are all present on the samples. How do we know which are active? My comment in the original review about the work of Hutchings et al. is unanswered.

Reviewer #3 (Remarks to the Author):

The authors have done a good job with the revision. This referee is satisfied with the changes and responses. The revised manuscript can now be accepted for publication.

Dear Prof. Bottari and Reviewers,

On behalf of my co-authors, we sincerely thank you and all reviewers for their valuable and constructive comments that we have used to improve the quality of our manuscript (ID: NCOMMS-17-00690A).

According to the referees' comments, a point-by-point response to the referees' questions was provided as shown in the following context. Our changes made to the manuscript are highlighted in red color. We hope that this revised manuscript will be acceptable for publication in your journal. Thank you very much for your consideration.

With best wishes,

Yanqiang Huang,
Dalian Institute of Chemical Physics,
457# Zhongshan Road, Dalian, 116023, China
Tel: +86-411-84379416
Fax: +86-411-84685940
E-mail: yqhuang@dicp.ac.cn

Responses to the Referees' comments

Referee: 1

Comments:

Most of the concerns have been addressed by the authors and the manuscript was significantly improved. Publication is recommended once the last issue is resolved.

The most important point the authors claim appears to be the new CO₂ reduction pathway based on the different reactivities of the Au/SiO₂-NH₂ and Au/SiO₂-Schiff materials. Since they have done some preliminary DFT studies, could they further show that the Schiff-base activated CO₂ is more reactive towards reduction (i.e. with a lower activation energy barrier)?

Response: Thanks for the comment. Following the reviewer's suggestion, we have simulated the chemisorption of CO₂ on the model of Au₅₅ sub-nanocluster (~1.2 nm) accommodated with an alkyl-imine Schiff-base. As seen in Supplementary Fig. 7, the gaseous CO₂ was found to be captured at the gold-Schiff base interface through a zwitterion intermediate, with an adsorption energy of 0.2–0.4 eV and its oxygen atoms bonded on the low-coordinated sites of gold nanocluster. The C-O stretch vibrational frequency was calculated to be in a range of 1701~1767 cm⁻¹ for various possible adsorption patterns, which was in agreement with the spectra observations.

We have also performed the mechanism study of CO₂ hydrogenation with DFT calculations based on our foregoing model of Au/SiO₂-Schiff catalyst under the COSMO solvent effect of water. As seen in Figure 1, the dissociation of H₂ to activated H species occurs on these low-coordinated corner sites of gold nanoclusters, with a barrier of 0.67 eV (TS-1) and a weak exothermic contribution (~0.10 eV). As mentioned above, the CO₂ molecule can be facilely captured as a zwitterion intermediate on the interface of Schiff-base and gold nanocluster, which can then be hydrogenated by the activated H to HCO₂ intermediate on gold surface, with a barrier of 1.00 eV (TS-2) and acting as the rate-determining step in the whole elementary

steps. By the help of another H adatom, the HCO₂ can be further, and easily hydrogenated to the cis-HCOOH on the catalyst surface by passing a barrier of 0.58 eV (TS-3), which subsequently desorbs from gold surface to get the desired trans-HCOOH product.

As for the Au/SiO₂-NH₂ catalyst, owing to that it experiences a proton transfer to the neighboring amine group, and form an ionic carbamate species, it is difficult for us to acquire the catalytic mechanism by the recent model of DFT calculations. While from our catalytic mechanism of Au/SiO₂-Schiff catalyst, the hydrogenation of zwitterion intermediate to HCO₂ intermediate behaves as the rate-determining step (1.00 eV, TS2), in which the breaking of N-CO₂ bond and the formation of H-CO₂ bond was considered as the main obstacles for transition. Moreover, the ionic carbamate was found to be thermodynamically more stable than the carbamate zwitterion, with a free energy for chemisorption of CO₂ (ΔG_{298}) about -8.89 kcal mol⁻¹ in water, and leads to a decrease of the distance between Lewis base center and CO₂ (~1.40 Å). As a result, it might also lead to a thermodynamic sink in CO₂ evolution, and we can imagine that a higher barrier will be required for the hydrogenation of ionic carbamate to formate. The corresponding discussion has been added in the revised manuscript (page 8 and 11).

Figure 1. A possible catalytic mechanism for CO₂ hydrogenation to formic acid over the Au/SiO₂-Schiff catalyst. Geometry with a closed packed Au₅₅ sub-nanocluster accommodated with an alkyl-imine Schiff-base under the COSMO solvent effect of water was used to model the real Au/SiO₂-Schiff catalyst in water. Energy profile was constructed based on the DFT calculation analysis of each elementary step.

Referee: 2

Comments:

The authors partially addressed my comments about wanting more experiment detail, and discussion of XPS and STEM, but in a very cursory fashion. The XPS shifts presumably tell us something about the state of the catalyst, but they only describe this in terms of the “electronegativity” of the gold. Similarly, the STEM results show that atoms, clusters, and nanoparticles are all present on the samples. How do we know which are active? My comment in the original review about the work of Hutchings et al. is unanswered.

Response: We are very sorry for misunderstanding the reviewer’s original comments. According to the XPS spectra of gold species in these two gold catalysts (Figure 2), the gold component of both were found to possess predominantly weak electronegativity and a low content of positive gold species, which were ascribed to the electron-donor capability of nitrogen groups toward the small gold clusters and some of isolated gold cations on the support, respectively. With DFT calculations (Figure 4 in the manuscript), it was found that the CO₂ molecule can be facilely captured as a zwitterion intermediate on the interface of Schiff-base and gold nanocluster, which can then be hydrogenated by the activated H to HCO₂ intermediate on gold surface, with a barrier of 1.00 eV (TS-2) and acting as the rate-determining step in the whole elementary steps. Therefore, the electron-rich gold surface caused by the electron donation from nitrogen groups might benefit to the hydrogenation of CO₂, since it could offer more negative hydride and lead to a higher reactivity of nucleophilic attack to the carbon center of CO₂ (*ACS Catal.*, 2017, 7, 4519–4526). It demonstrated that the gold species with lower binding energy is one of factors governing the high activity of the functionalized gold catalysts. The corresponding discussion can be found in the revised manuscript (Page 12).

Figure 2. XPS spectra of the Au 4f core levels of the as-prepared gold catalysts. (a) Au/SiO₂-NH₂, **(b)** Au/SiO₂-Schiff and, **(c)** unfunctionalized Au/SiO₂. Besides metallic gold, both the functionalized samples contained some form of cationic Au species that were ascribed to sub-nanometric cationic Au clusters and isolated Au atoms.

For further identifying the active gold species among the Au/SiO₂-Schiff, another two Schiff-base modified gold catalysts, with one of a larger metal particle size (~3.2 nm) and the other containing only the single-atom gold species, were also prepared by the surface-functional group promoted *in situ* reduction method and the milling-mediated solid reduction method, respectively (Figure 3 and Figure 4), and were then evaluated at the same reaction conditions. However, the superior activity of Au/SiO₂-Schiff catalyst was greatly reduced or even quenched (entry 13 and 14 of Table 2 in the manuscript). It suggests that the hydrogenation of CO₂ over gold nanocatalysts exhibits a great size-dependent behavior, with the small sub-nanoclusters more efficient than the large-sized nanoparticles, but the single-atom gold species were ineffective as catalyst in the catalytic hydrogenation of CO₂. The similar size-dependency of catalytic activity was also reported previously for the CO oxidation over Au/FeO_x materials (Science, 2008, 321, 1331–1335). The corresponding discussion has been added in the revised manuscript (page 7). Thanks for your insight.

Figure 3. Typical HAADF-STEM images of the single-atom Au/SiO₂-Schiff catalyst. A novel milling-mediated solid reduction method was developed to prepare the single-atom Au catalyst (for details, please see *Sample preparation* in the manuscript). During grinding, the gold precursor was *in situ* reduced by the spillover hydrogen from NaBH₄ in solid state. This solvent-free approach was benefit to restrain crystal nucleation and give a single-atom catalyst with low metal loadings (~0.1 wt%). As shown in the figures, the Au species are mainly dispersed as single atoms (white squares), as well as some negligible nanoclusters (white circle).

Figure 4. Typical HAADF-STEM images of Au/SiO₂-Schiff catalyst prepared by *in situ* reduction method. The figures indicated that both the nanoparticles (larger than 2 nm) and individual Au atoms (indicated by white squares) are present in the inactive catalyst.

Referee: 3

Comments:

The authors have done a good job with the revision. This referee is satisfied with the changes and responses. The revised manuscript can now be accepted for publication.

Response: We appreciate your recommendation of acceptance and helpful comments in the reviewing process.

Reviewer #1 (Remarks to the Author):

I have carefully gone through the comments and replies. I believe all the issues were fully addressed and the publication is recommended.